# Post-transcriptional 3′-UTR cleavage of mRNA transcripts generates thousands of stable uncapped autonomous RNA fragments

Yuval Malka[1], Avital Steiman-Shimony[2], Eran Rosenthal [3], Liron Argaman[2], Leonor Cohen-Daniel[1], Eliran Arbib[1], Hanah Margalit [2], Tommy Kaplan [3] & Michael Berger [1]

The majority of mammalian genes contain one or more alternative polyadenylation sites. Choice of polyadenylation sites was suggested as one of the underlying mechanisms for generating longer/shorter transcript isoforms. Here, we demonstrate that mature mRNA transcripts can undergo additional cleavage and polyadenylation at a proximal internal site in the 3′-UTR, resulting in two stable, autonomous, RNA fragments: a coding sequence with a shorter 3′-UTR (body) and an uncapped 3′-UTR sequence downstream of the cleavage point (tail). Analyses of the human transcriptome has revealed thousands of such cleavage positions, suggesting a widespread post-transcriptional phenomenon producing thousands of stable 3′-UTR RNA tails that exist alongside their transcripts of origin. By analyzing the impact of microRNAs, we observed a significantly stronger effect for microRNA regulation at the body compared to the tail fragments. Our findings open a variety of future research prospects and call for a new perspective on 3′-UTR-dependent gene regulation.

[1] The Lautenberg Center for Immunology and Cancer Research, IMRIC, Faculty of Medicine, The Hebrew University, Jerusalem 9112001, Israel. [2] Department of Microbiology and Molecular Genetics, IMRIC, Faculty of Medicine, The Hebrew University, Jerusalem 9112001, Israel. [3] School of Computer Science and Engineering, The Hebrew University, Jerusalem 9190401, Israel. Correspondence and requests for materials should be addressed to Y.M. (email: yuvalmalka79@gmail.com) or to T.K. (email: tommy@cs.huji.ac.il) or to M.B. (email: michaelb@ekmd.huji.ac.il)

The majority of mammalian genes undergo alternative polyadenylation (APA) in the nucleus to generate RNA isoforms with shorter 3′-UTRs. Accumulation of transcripts with short 3′-UTRs is widespread across tissues and is positively correlated with cell proliferation[1], membrane protein localization[2], and several diseases including cancer[3–6].

3′-UTRs serve as major docking platforms for regulatory proteins and RNAs, including microRNAs (miRNAs). Hence, it was initially assumed that APA increases the stability of the mRNAs and thus increases protein output[7]. However, recent studies have demonstrated that the choice of APA sites has minimal effects on stability and translational efficiency[8, 9]. Thus, the functional significance of this regulation has remained poorly understood.

Recent results showed that mRNAs from defined neuron populations exhibit widespread unbalanced expression of cognate 3′-UTRs relative to respective upstream coding sequences (CDSs)[10]. The presence of a capped 3′-UTR transcript separately from its associated CDS has also been documented by Mercer et al.[11] for some genes in humans, mice, and flies. Despite the potential high impact of these results on the current understanding of the transcriptome, no large-scale study of this phenomenon has been yet undertaken.

In this study, we show that thousands of mature poly(A) mRNA transcripts could often be cleaved post-transcriptionally into two fragments: a "body" unit found upstream of the cleavage site and a downstream uncapped "tail" unit. By re-annotating the human transcriptome for novel cleavage positions, we show stronger regulatory effects by miRNA sites in the body fragment compared to 3′-UTR sites downstream of the cleavage point.

## Results

**3′-UTR cleavage results in autonomous uncapped RNA fragments**. To systematically elucidate the process of 3′-UTR shortening, we turned to analyze published mRNA sequencing (RNA-seq) data from naive mouse T cells, where 3′-UTR shortening is widespread and associated with the transition of the cells from the naive to activated state[7]. Manual examination of RNA-seq data[12] revealed many genes with a distinctive gap in their read coverage along the 3′-UTR, adjacent to known APA sites[13] (Fig. 1a and Supplementary Fig. 1). Moreover, these gaps were also frequently observed in published RNA-seq data from other tissues and cells[14, 15] (Fig. 1a and Supplementary Fig. 2a–o). These gaps are characterized by unbalanced coverage of sequenced reads at their two sides, especially in brain tissue, suggesting that the gap may represent a cleavage site that divides the transcript into two independent fragments with different expression patterns. For example, Fig. 1a illustrates the RNA-seq read coverage in brain tissue for three genes: *Ssr1* presents higher read coverage downstream of the gap site; *Bcl2* shows higher read coverage upstream of the gap site; and *Rab2a* demonstrates a similar coverage on either side.

To determine whether the observed gap represents an authentic cleavage within the transcript, we used quantitative real-time PCR (qRT-PCR) to compare the relative expression levels of the regions upstream, downstream, and across the gap of *Ssr1*, *Bcl2*, and *Rab2a* genes. Initially, total RNA extraction showed no substantial or minor change of expression between the three amplicons (Supplementary Fig. 3). However, comparison of the relative expression of each region (upstream, downstream, and across the gap) between the cytoplasm and the nucleus, showed a significant decrease of the amplicons corresponding to the gap region in the cytoplasm, while the upstream and downstream amplicons demonstrated similar expression levels (Fig. 1b). These results indicate that in the cytoplasm most of the

observed transcripts are assembled from two independent RNA fragments, while in the nucleus these transcripts seem to originate from one long transcript. In addition, these results complement the mRNA-seq results by demonstrating that the gap region indeed represents a cleavage point.

These findings support a model by which some transcripts could be cleaved at APA sites post-transcriptionally, resulting in two autonomous RNA molecules. We therefore hypothesized that the downstream fragment should be uncapped at its 5′ end (Fig. 1c). To test this hypothesis, RNA extracts from both the nucleus and cytoplasm were treated with terminator 5′-phosphate-dependent exonuclease (TEX) that degrades uncapped RNA molecules. We then used qRT-PCR to quantify the levels of corresponding upstream and downstream fragments. As Fig. 1d shows, TEX treatment of RNA results in a sharp decrease in cytoplasmic levels of the fragments downstream of the putative cleavage point (compared to their "body" fragment) with limited changes in the nucleus ($p > 0.01$). To further substantiate these results we performed Northern blot analysis on Actb transcript (a highly expressed gene with high mRNA-seq coverage and a substantial gap at the 3′-UTR) using probes for either the gene body or the tail (Supplementary Fig. 4a, b). Analysis using a body probe indicates that most Actb copies are found in the short (cleaved) form and only a few copies in the canonical (uncleaved) form. Both isoforms are TEX-insensitive. Conversely, the tail probes identified both the canonical and tail fragments in untreated RNA, with only the former present following TEX treatment. Overall, the results of the qRT-PCR and Northern blot experiments following TEX treatment indicate that mRNA can be cleaved at 3′-UTR into (1) the gene body, containing the CDS with a shortened 3′-UTR and (2) one or more cleaved fragments, containing the downstream "tails" of the canonical transcript. These tail fragments are sensitive to TEX treatment, indicating they are uncapped. Importantly, we observe higher sensitivity of the tail fragments to TEX (compared to their body fragment) in the cytoplasm compared to the nucleus, indicating that these fragments are overrepresented in the cytoplasm in comparison to the nucleus. These results raise the possibility that cleavage occurs post-transcriptionally, potentially by the canonical cleavage and polyadenylation machinery.

Finally, to further support the notion that tail fragments can be separated from the gene body, and to map the start site of the 3′-UTR tail fragment at a single nucleotide resolution, we performed 5′-Rapid Amplification of cDNA Ends (5′-RACE) for tail fragments of six selected genes that presented a substantial gap in APA site according to published RNA-seq data (Supplementary Figs. 4c and 5a–f). Sequencing of the 5′-RACE amplified products identified cleaved 3′-UTR fragments adjacently to APA sites in all six samples. These results further support our model that 3′-UTR tails can be separated from their gene bodies, likely through an APA mechanism.

**Analysis of 3′-UTRs identifies thousands of autonomous RNA tails**. In the following sections, we describe three large-scale experiments to further support the suggested model at a transcriptome-wide scale.

We developed three high-throughput methods to distinguish between capped body fragments and uncapped tail fragments at a transcriptome-wide scale (Fig. 2a). On the one hand, to enrich for capped body fragments, poly(A) selected RNA from U2OS cells was either treated with TEX to degrade the uncapped tail fragments or immunoprecipitated (IP) using anti-CAP antibody (CAP IP). On the other hand, to enrich for tail fragments, poly(A) selected RNA was subjected to an in vitro capping system with biotinylated guanosine triphosphate (GTP) followed by pulldown

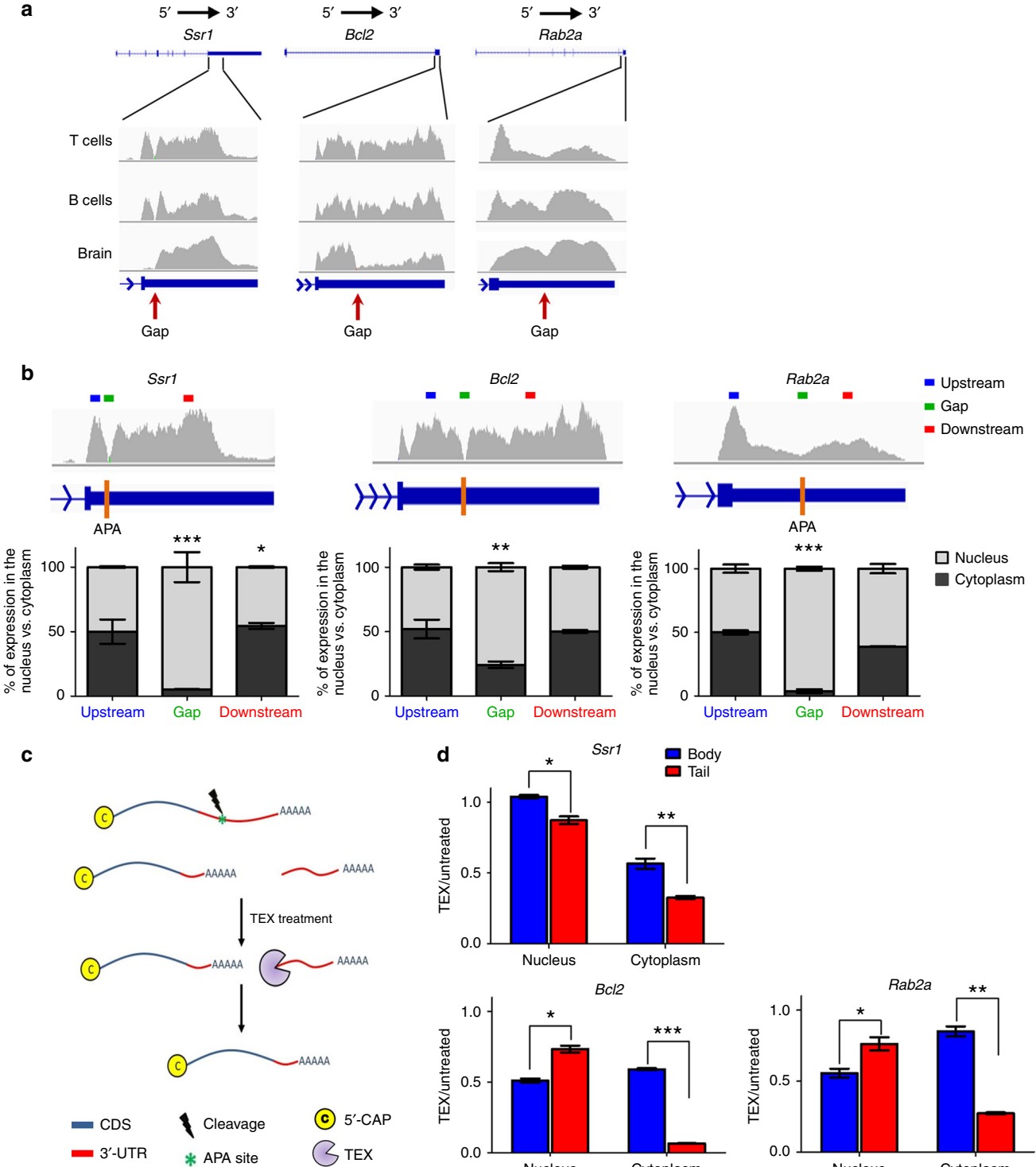

**Fig. 1** Cleavage of 3′-UTR regions results in autonomous uncapped RNA fragments. **a** RNA-seq read coverage (y-axis) for *Ssr1*, *Bcl2*, and *Rab2a* in mouse T cells[12], B cells[14], and brain tissue[15], demonstrating a gap in read coverage (marked by arrows) at 3′-UTRs, as well as uneven levels of RNA upstream and downstream of the gap site. **b** Three sets of primers were used per gene for qRT-PCR amplification: upstream (blue rectangle), downstream (red), and across the RNA-seq coverage gap (green). Bar plots (bottom) visualize the relative percentage of cytoplasmic (black) and nuclear (gray) expression of each amplicon out of its total (cytoplasm+nuclear) expression. **c** A model for post-transcriptional processing of mRNA: mRNA is cleaved at an APA site, resulting in a capped "body" and an uncapped 3′-UTR "tail," sensitive to terminator 5′-phosphate-dependent exonuclease (TEX) treatment. **d** qRT-PCR analysis shows cytoplasmic degradation of the "tail" unit (in red) compared to the "body" unit (in blue) following TEX treatment. *$p < 0.05$, **$p < 0.01$ and ***$p < 0.001$ (two-tailed Student's t-test). Results are representative of three independent experiments (**b**, **d**). Error bars = s.d.

(PD) of the newly capped fragments (3′-PD) using streptavidin beads. The RNA from these three assays was validated by qRT-PCR on four selected genes that presented a substantial gap in APA site according to published RNA-seq data (Supplementary Fig. 6a–c), and analyzed by whole-transcriptome RNA-seq (see

representative results in Fig. 2b). To identify a possible cleavage point for each transcript/treatment, we developed a computational model that compares the overall normalized read coverage along the transcript, and seeks for a transition point between body and tail coverages. This was done by first normalizing the

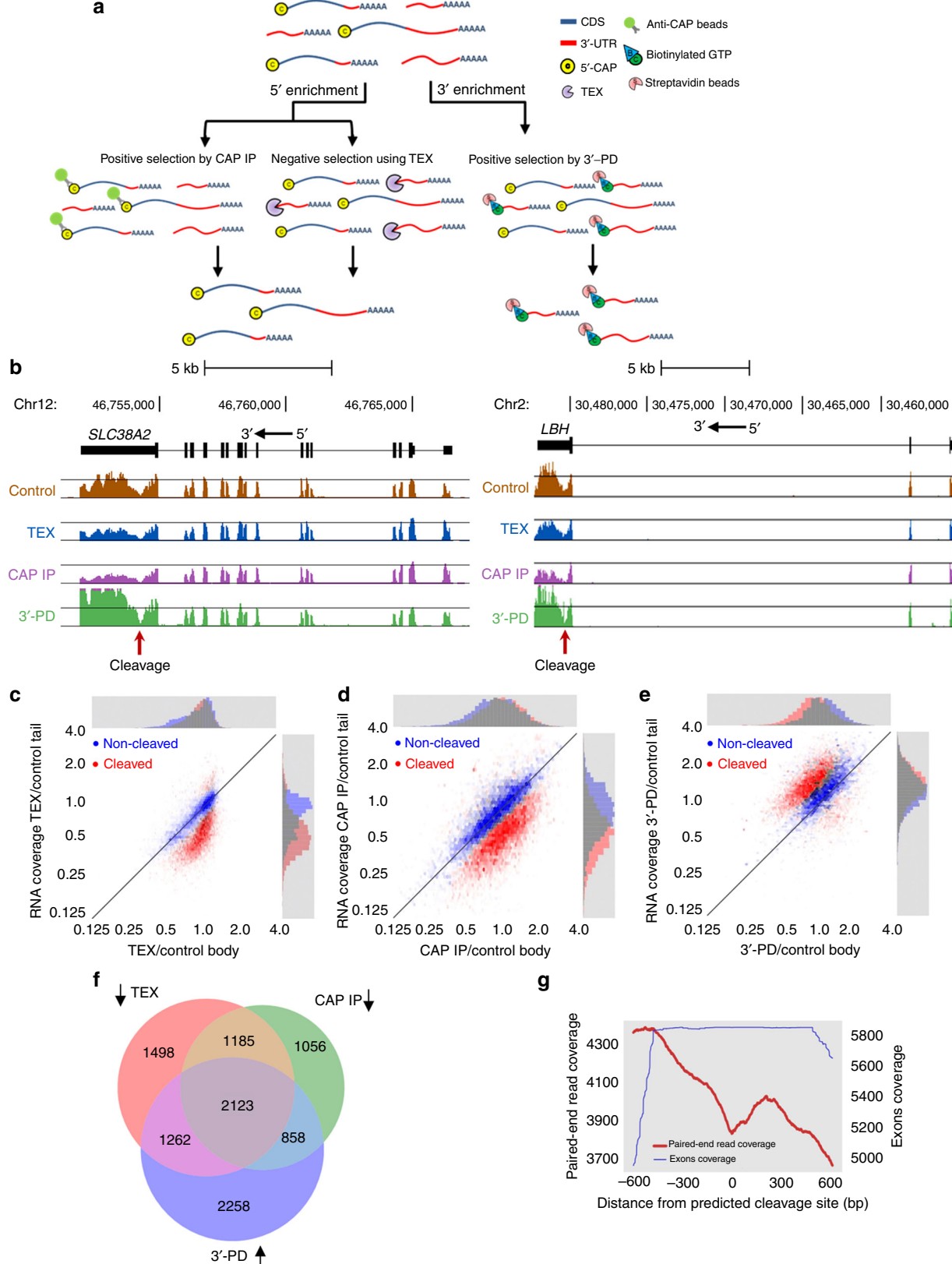

transcriptome-wide data from each treatment, and then using a Gaussian Hidden Markov Model (HMM) with two states, corresponding to the body and tail regions. Then, using the Viterbi algorithm we identified for each transcript the most probable transition point and used Kolmogorov–Smirnov test to determine its statistical significance (see Methods and Supplementary Fig. 7a–e). Overall, we identified 6068 genes (out of 17,393 genes analyzed) with a statistically significant cleavage in the 3′-UTR, following TEX treatment (false discovery rate (FDR) < 0.01) and 5222 genes in CAP IP experiment (out of 17,433 genes), with an overlap of 3308 genes (55–63% of the genes in either group). Most of these genes also show high agreement in the location of the predicted cleavage sites within different data sets (66% within 350 nt; Supplementary Fig. 8a–e). In addition, we identified 6501 genes (out of 17,023 tested) with a significant increase of the tail read coverage in 3′-PD treatment (FDR < 0.01). Overall, in all three treatments we report 2123 common genes showing a significant coordinated change in read coverage, with an average decrease of 34% in tail vs. body coverage for TEX and CAP IP experiments, and 41% increase in tails for the 3′-PD experiment (Fig. 2c–f and Supplementary Data 1–3). These results suggest that for thousands of genes, stable cleaved transcript tails are present in the cell in substantial amounts.

To further validate these findings, we analyzed the untreated poly(A) selected RNA (control), and calculated the average read coverage across the putative cleavage points predicted by our computational model based on TEX-treated RNA. The predicted cleavage points were characterized by well-positioned troughs of low read coverage compared to the surrounding regions (Fig. 2g, red line), in agreement with results shown in Fig. 1. Further analysis revealed that for most genes (>60%), the putative cleavage positions are located within 500 nt of the stop codon (Supplementary Fig. 8f), suggesting a relatively short 3′-UTR region. Similar results were obtained using TEX-treated RNA from HEK-293 cells (6574 genes with predicted cleavage point out of 17,506 analyzed; Supplementary Fig. 9a–c and Supplementary Data 4). Comparison of TEX treatment between HEK-293 and U2OS cells demonstrates an overlap of 3432 cleaved genes (52–57%; Supplementary Fig. 9d).

### 3′-UTR processing is post-transcriptional and APA-dependent.
Based on our initial observations that cleavage sites are associated with APA sites, we postulated that the processing mechanism is APA-dependent.

To test this hypothesis, we compared the cleavage points as inferred by the HMM model following TEX treatment in HEK-293 cells to the polyadenylation sites documented in HEK-293 cells in the APA repository PolyASite[13]. Our results, as seen in Fig. 3a, demonstrate that the probability of observing an APA site peaks at the HMM predicted cleavage point, and is significantly higher compared to random positions (Kolmogorov–Smirnov test, p < 2e−7).

To substantiate the APA dependency of the cleavage sites, we mutated the polyadenylation signal (PAS) motifs in the 3′-UTRs of two genes (ST6GALNAC4 and RAP1B). 3′-UTR of wild-type (WT) and mutated PAS were cloned into pGL3 luciferase reporter constructs. qRT-PCR analysis of cytoplasmic RNA fraction treated with TEX showed significant reduction in the WT tail fragments but not in the tail fragments of the mutated PAS (Fig. 3b, c). In addition, mutation of the PAS site restored the relative RNA levels at the APA site (without TEX treatment, Supplementary Fig. 10a–d).

For an additional evaluation of our model and to quantitatively map RNA 3′ ends, we turned to perform 3′-end RNA-seq in untreated and TEX-treated cells. According to our model, cleaved genes should present two 3′-end RNA-seq peaks in the 3′-UTR: a proximal peak corresponding to the shortened body fragment and a distal peak that represents the 3′ ends of both the canonical and cleaved tail. Following TEX treatment, we expect a decrease in the height of the distal peak (due to the TEX sensitivity of the cleaved tail) compared to the proximal peak (TEX-insensitive body). Indeed, we observed a dramatic decrease in the relative height of the distal peak (normalized to the proximal peak) following TEX treatment (−37% and −56.5% for DDX21 and PRKCA, respectively; Fig. 3d and Supplementary Fig. 11). Overall, we observed an average of 22% decrease in distal vs. proximal peak heights (p < 4.5e−15, Fig. 3e and Supplementary Data 5), in accord with our previous estimation of the overall cleaved tail population, based on the decrease in tail vs. body RNA-seq coverage following TEX treatment (27% decreases; Fig. 2c).

To substantiate that RNA cleavage occurs post-transcriptionally, we repeated the 3′-end RNA-seq and qRT-PCR experiments in cells treated with α-amanitin, an RNA polymerase II-specific and III-specific inhibitor. qRT-PCR analysis for several genes following TEX treatment demonstrated a strong decrease in the relative expression of the tail fragment, but not the body fragment, 6–9 h after α-amanitin treatment (compared to TEX-untreated RNA in α-amanitin-treated cells; Fig. 3f and Supplementary Fig. 12a, b). This suggests that cleavage is not necessarily coupled with active transcription, and cannot be explained by differences in body/tail stability.

In line with these results, as we show for STX16 (Fig. 3g), treatment by α-amanitin for 9 h allows for additional processing of the canonical transcript, and is reflected by a stronger (proximal) 3′-end RNA-seq peak at the cleaved body compared to untreated cells. This trend is further emphasized in following TEX treatment (Fig. 3g, bottom).

**Fig. 2** Transcriptome-wide analysis of 3′-UTR regions identifies thousands of stable cleaved tails. **a** We used three experimental methods to measure separated body and tail RNA fragments at a transcriptome-wide scale; poly(A) selected RNA from U2OS cells was enriched for 5′-capped bodies using TEX treatment (TEX) or anti-Cap immunoprecipitation (CAP IP); uncapped tails were enriched by streptavidin bead pulldown of in vitro biotinylated-7-methylguanylate capped RNA (3′-PD). **b** RNA-seq read coverage data (y-axis) across the putative cleavage point of SLC38A2 and LBH (arrow) show reduced "tail" read coverage in TEX- and CAP IP-treated cells compared to enriched "tails" in 3′-PD, and equal coverage in untreated RNA (control). Black horizontal lines mark average coverage across exons. **c–e** Computational analysis using a Hidden Markov Model (HMM) was applied to identify the most probable cleavage point for each transcript, resulting in 12,578 statistically significant transcripts (Kolmogorov–Smirnov p < 0.01) for TEX (overall of 6068 cleaved genes, FDR < 0.01; 11,108 transcripts for CAP IP (5222 genes); and 14,589 transcripts for 3′-PD (6501 genes). Shown are density plots comparing the relative read coverage (normalized over coding region, genome wide) before and after treatment, for bodies (x-axis) and tails (y-axis) for TEX (**c**), CAP IP (**d**), and 3′-PD (**e**). Red dots correspond to transcripts predicted to be cleaved, and blue dots mark non-cleaved transcripts. Histograms above and to the right of each plot show the marginal distribution of body or tail treated to untreated RNA-seq ratio for each population. **f** Venn diagram of genes with statistically significant differences in body vs. tail read coverage (FDR < 0.01) for TEX, CAP IP, and 3′-PD. **g** Meta-gene analysis of average read coverage (paired-end RNA-seq; red line) showing a dip at the putative cleavage point (predicted in TEX treatment data). Blue line shows meta-gene plot of exon annotations, suggesting that the observed dip is not due to exon–intron junctions

Transcriptome-wide analysis of the 3′-end RNA-seq data showed a significant decrease ($p < 1.8\text{e}{-}17$) in the distal peak height compared to the proximal peak following TEX treatment in α-amanitin-treated cells (Fig. 3h, Supplementary Figs. 13a, b, and Supplementary Data 5) and this RNA processing seem to continue post-transcriptionally (Supplementary Fig. 13c).

Finally, we compared the location of the putative cleavage site (as predicted by the HMM model based on TEX data) with the

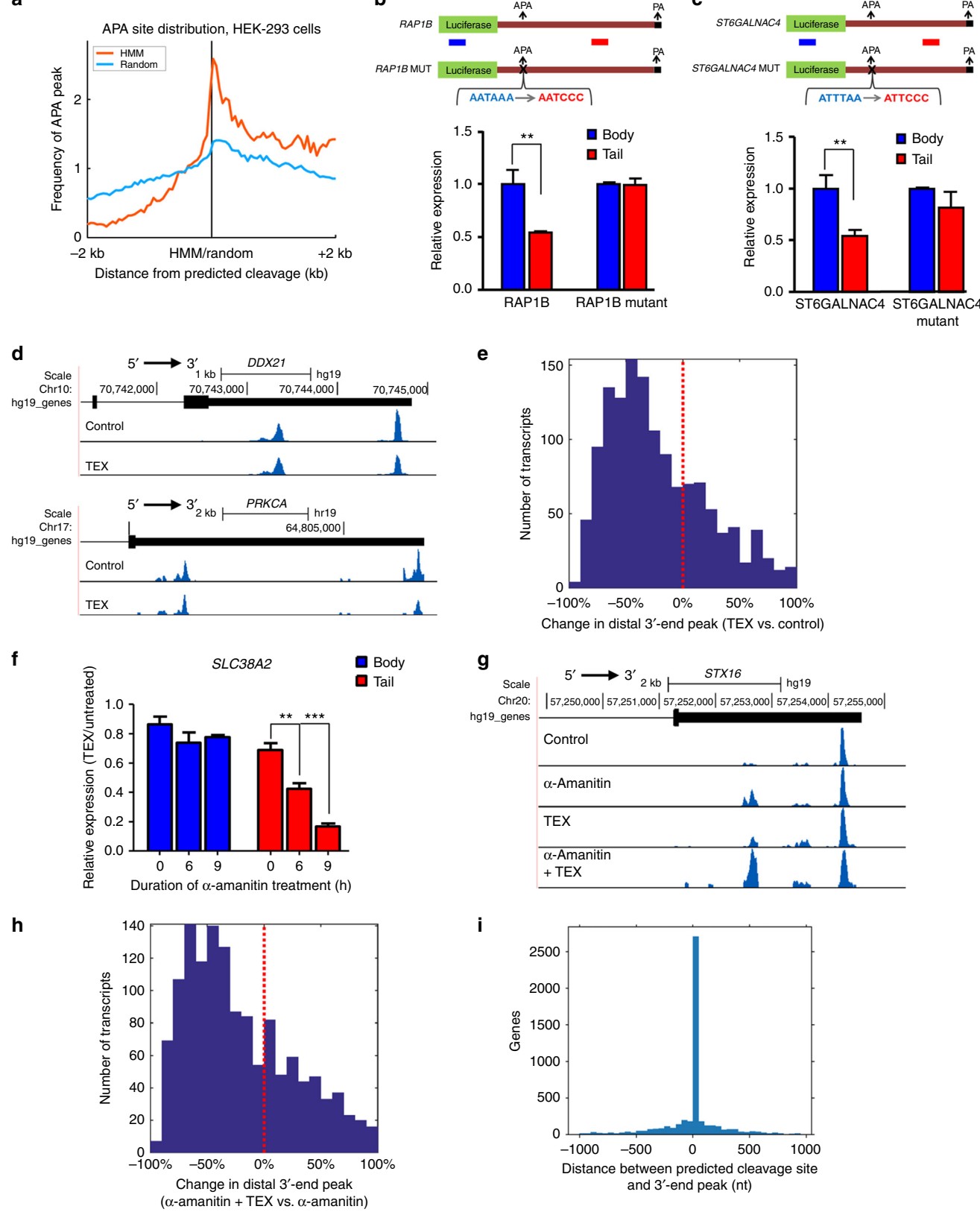

nearest 3′-end RNA-seq peak (APA site) in the same cell type (U2OS). For 80% of the cleaved transcripts, the predicted cleavage site appeared within 100 nt from the most proximal 3′-end RNA-seq peak (Fig. 3i). These results strongly suggest that the cleavage point occurs at APA sites.

**Body and tail RNAs are independently regulated by miRNAs.** To corroborate that the cleavage generates two independent transcripts, we tested whether the transcript tail and body were affected differently by miRNAs. To this end, we conducted transfection experiments of two miRNAs, hsa-miR-92a-3p and hsa-let-7a-5p, in HEK-293 cells (Fig. 4a). The two miRNAs are known to be expressed in HEK-293 cells, with many validated targets. While the conventional analysis would predict that the complete transcript would be down-regulated in a uniform manner (in terms of RNA-seq coverage), we expected that if the binding site of the miRNA was located at the transcript body, there would be a stronger decrease in the expression level of the transcript body compared to its tail (Fig. 4b). We annotated the body and tail regions of transcripts using the putative cleavage points predicted by our HMM model from HEK-293 cells treated with TEX, as described above. By separately analyzing the change in expression along the gene body and tail, we found that the set of transcript bodies that contain a putative miRNA binding site showed a statistically significant larger decrease in expression compared to the set of tail regions ($p \leq 1.2\mathrm{e}{-}11$, Fig. 4c and Supplementary Fig. 14a; $p \leq .2.8\mathrm{e}{-}10$, Supplementary Figs. 14b, c, and Supplementary Data 6 and 7).

Interestingly, the conventional analysis of the entire transcripts showed a smaller change in expression than that of the transcript body ($p \leq 9.4\mathrm{e}{-}7$, see Fig. 4c, green vs. blue lines and Supplementary Fig. 14a green vs. blue boxplot; Supplementary Fig. 14b, $p \leq 2.2\mathrm{e}{-}16$, and Supplementary Fig. 14c; and representative examples in Fig. 4d). These results suggest that conventional analysis of the entire transcript (which average the effect of possibly separated body and tail fragments) might undervalue the overall effect of miRNA regulation.

To verify that the different effects of the miRNA observed for body and tail are associated with the cleavage position, we conducted a similar analysis comparing the effect of the miRNA on CDS vs. 3′-UTR in transcripts that were not predicted to have cleavage sites. As shown in Fig. 4e, the difference between the expression fold change (FC, shown in $\log_2$ scale) between the CDS and the 3′-UTR was significantly smaller than the difference between the body and tail ($p \leq 7.9\mathrm{e}{-}26$ in transcripts with sites in either body/CDS or in tail/3′-UTR). The same was demonstrated for let-7a transfection (Supplementary Fig. 14d). Analysis of the data for miRNAs with binding sites in the tail region was inconclusive (Supplementary Fig. 14e–h). As most mammalian genes are known to have more than one APA site, several

cleavages may occur in one transcript. Our analysis was based on the first cleavage point found computationally following TEX treatment, hence while the body fragment is clearly defined, it is difficult to predict the boundaries of the tail fragment and whether or not the predicted miRNA binding site is included within it.

Our findings demonstrate that many mRNA transcripts with canonical polyadenylation sites are processed post-transcriptionally through an alternative cleavage and polyadenylation mechanism. As a result of this process, the downstream cleaved fragment of the 3′-UTR remains uncapped and stable, and is maintained alongside its corresponding CDS as an independent entity. Since many RNAs are composed of more than one fragment, unbalanced stabilization kinetics between the different fragments under various biological conditions may favor abundance of one over the other. Hence, our discovery sheds a new light on biological processes such as RNA shortening[7] and expression of independent 3′-UTR fragments[10, 11]. Moreover, separate analysis of the tail and body enables the identification of more miRNA targets (Fig. 4f).

## Discussion

Recently, the presence of 3′-UTR transcripts, separated from their associated mRNAs, was documented for some genes[10, 11]. Although these results may affect our current understanding of the transcriptome, a large-scale study of this phenomenon has not yet been performed.

Here, using a variety of molecular and genomic methods, we now show that thousands of mature poly(A) mRNAs, located mainly in the cytoplasm, are cleaved and polyadenylated post-transcriptionally at APA sites in their 3′-UTRs, and regions that were considered to be a part of the canonical transcripts are in fact autonomous uncapped "tail" units. Most likely, this wide-spread phenomenon, which occurs in thousands of genes, was overlooked due to the stable nature of the tails that makes them hard to detect using next-generation RNA-seq data. To overcome this difficulty, we systematically enriched for body or tail units based on the presence of 5′-cap modifications, followed by deep RNA-seq.

By analyzing these data we re-annotated the human transcriptome for novel cell type-specific cleavage positions, and identified thousands of 3′-UTR RNA tails that exist alongside their transcripts of origin. We further established our results by explicitly mapping transcript ends using 3′-end RNA-seq, and related the cleavage locations to known APA sites. These results indicate that in addition to termination and polyadenylation at the canonical polyadenylation site, additional cleavage and polyadenylation occurs within many transcripts. Treating these cells with the transcription inhibitor α-amanitin resulted in the accumulation of both "body" and "tail" units, suggesting that this

**Fig. 3** 3′-UTR processing is APA-dependent and occurs post-transcriptionally. **a** Distribution of HEK-293 polyadenylation sites (from PolyASite[13]) surrounding the predicted cleavage point (HMM, in orange line) or random points in random UTRs (blue line). **b, c** Mutations in APA sites diminish cleavage. Mutation of 3′-UTR APA sites cloned into a pGL3 promoter luciferase reporter system shows reduced cytoplasmic sensitivity to TEX in *RAP1B* (**b**) and *ST6GALNAC4* (**c**). **d** 3′-end RNA-seq data (*y*-axis) for *DDX21* and *PRKCA* show relative decreases in the distal peak (corresponding to the 3′ ends of both canonical and "tail" transcripts) compared to the proximal peak (corresponding to the 3′ end of the "body" fragment) following TEX treatment (1484 transcripts). In total, 65% of the distal peaks are reduced following TEX treatment ($p \leq 4.5\mathrm{e}{-}15$ using a paired *t*-test on distal/proximal ratio), with an average decrease of 22% in the distal peak height following TEX. **e** Histogram showing the relative change of distal peaks height (3′-end RNA-seq) following and before TEX treatment. **f** 3′-UTR cleavage occurs post-transcriptionally. RT-qPCR in U2OS cells shows strong TEX sensitivity in the cytoplasmic *SLC38A2* "tail" fragment 6 or 9 h following α-amanitin treatment compared to TEX-untreated RNA from α-amanitin-treated cells. **g** 3′-end RNA-seq of *STX16* in control, following α-amanitin; TEX treatment; and both α-amanitin and TEX, show an increase in proximal peak. **h** Same as **e**, for α-amanitin-treated cells for 9 h, after and before TEX treatment ($p \leq 1.8\mathrm{e}{-}17$, paired *t*-test). **i** Histogram of distances between predicted cleavage sites (HMM model using TEX data) and nearest 3′-end RNA-seq peak (in 5619 transcripts). **p < 0.01 and ***p < 0.001 (two-tailed Student's *t*-test). Results are representative of three independent experiments with triplicates (**b**, **c**, and **f**). Error bars (**b**, **c**, and **f**), s.d.

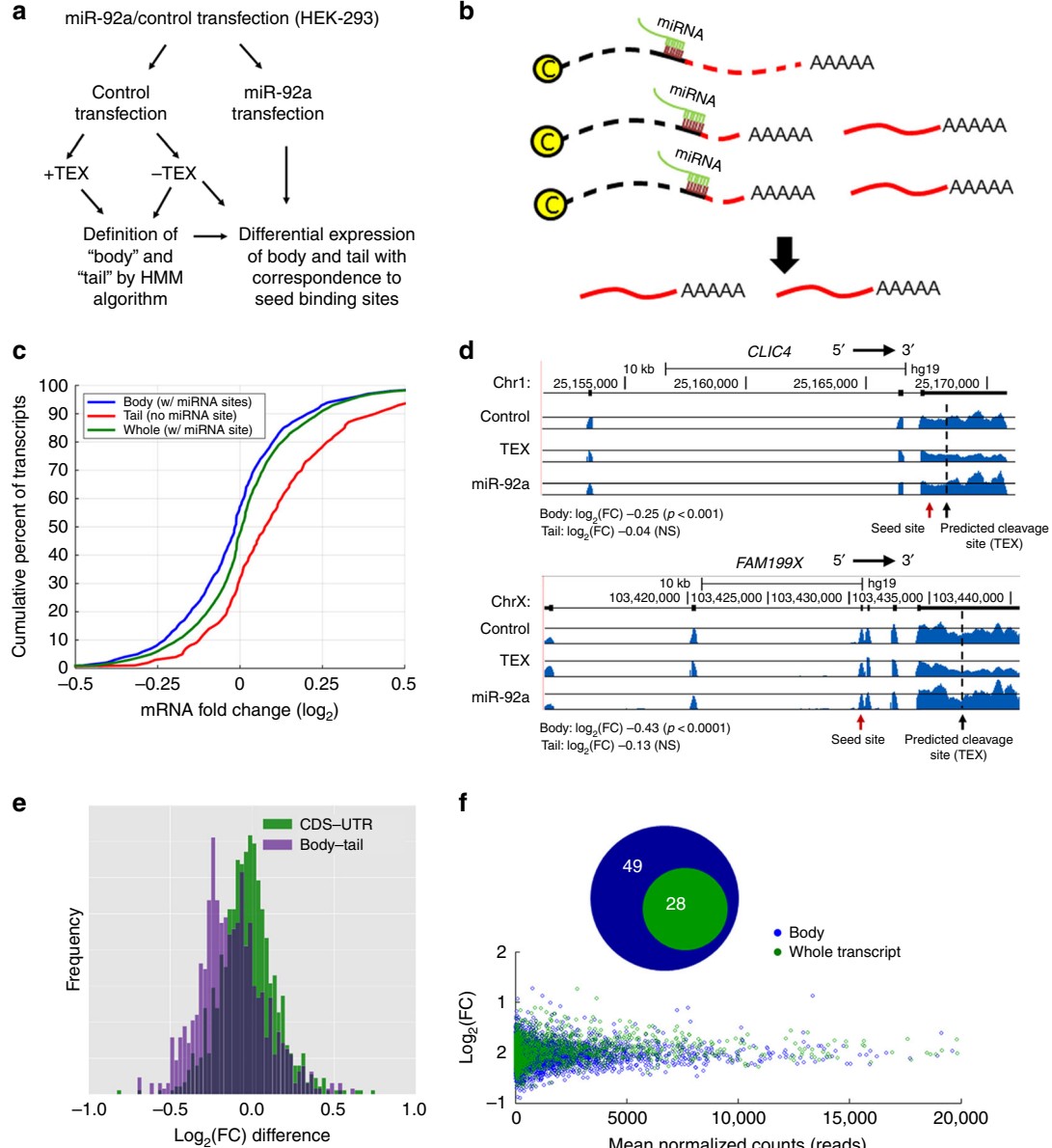

**Fig. 4** Body and tail regions of cleaved transcripts are independently regulated by miRNAs. HEK-293 cells were transfected with synthetic miR-92a-3p or with a control double-stranded RNA (dsRNA), and total RNA was extracted 40 h after transfection. **a** Outline of experiment and data analysis. **b** Simplified depiction of expected results: down-regulation of transcripts with body miRNA binding sites, independent of their tails. **c** Cumulative frequency distributions of expression changes in transcripts with 7-mer seed binding sites in the body. *X*-axis corresponds to log$_2$ of expression fold change (log$_2$(FC)). Comparison was conducted by a two-sided Kolmogorov–Smirnov test. **d** Normalized RNA-seq read counts from the control, TEX-treated RNA, and miR-92a-transfected cells for *CLIC4* (top) and *FAM199X* (bottom). Predicted cleavage sites and miR-92a 7-mer seed sites in their body region are shown (arrows). **e** Histograms of log$_2$ fold change differences in body vs. tail regions (purple) in transcripts with predicted cleavage site; or between coding sequence (CDS) vs. 3′-UTR (light green) in transcripts without predicted cleavage site; and the overlap between them (dark purple). In all transcripts, a miR-92a binding site was located exclusively in either body/CDS region, or in tail/3′-UTR region. **f** Body/tail distinction identifies additional miRNA targets. Number of transcripts with statistically significant decrease in expression following miRNA overexpression based on the custom annotations (blue) and based on the conventional whole transcript annotation (green). Transcripts contain a 7-mer binding site in the body region. (Bottom) A scatter plot of the mean normalized counts of transcripts and their log$_2$ fold change by the body–tail annotation (blue plots) and by the conventional whole transcript annotation (green plot)

3′-UTR cleavage is transcriptionally independent and involves the processing of mature mRNA.

Our data suggest that cleaved RNAs are present in the cytoplasm more frequently than in the nucleus. These findings are supported by Furger and colleagues[16] who described that APA isoforms with shorter 3′-UTRs are more frequent in the cytoplasm. In addition, although it is accepted that cleavage and polyadenylation occurs co-transcriptionally, our study as well as

others[17, 18] demonstrates that cleavage and polyadenylation at APA sites may occur post-transcriptionally as well. These two observations could be explained by mRNA retention in the nucleus followed by additional cleavage and export to the cytoplasm. Indeed, cleavage and polyadenylation in the nucleus are presented in Prasanth et al.[17], describing that some transcripts (such as *CTN-RNA*) are retained in the nucleus and are cleaved post-transcriptionally upon stress conditions to produce new

mRNA transcripts with shorter 3′-UTRs. In addition, Jenal et al.[18] synthesized RNA fragments with WT or mutated PAS proximal APA sites and then incubated with nuclear extract of HeLa cells. In this in vitro assay, cleavage occurred at RNA fragments containing WT APA but not in RNA fragments containing mutated PAS[18]. Additionally, our observations could be also explained by cleavage and polyadenylation at APA sites in the cytoplasm. However, to date, there is no evidence of such mechanisms.

While the observation and characterization of thousands of uncapped tail fragments raise an intriguing question of their biological function and importance, we cannot overrule the possibility that some of these fragments are byproduct results from APA usage due to a fail-safe mechanism ensuring full transcription and correct splicing are obtained to produce mature 3′ ends[19].

miRNAs are known to regulate mRNA transcripts by binding primarily to the 3′-UTRs. Hence, we assessed the impact of our findings by evaluating the effect of the transfection of a given miRNA on its targets by evaluating the expression of the "body" and "tail" fragments with binding sites independently rather than the entire transcript, as has been done in conventional analyses. Our analysis demonstrated that following miRNA over-expression, the entire transcript shows a significantly smaller change in expression compared to the transcript body. We find this to be especially significant, as in many cases transcripts that were not found to be down-regulated significantly by the "conventional" analysis of the full transcript were found to be down-regulated when only the body was considered. Previous studies demonstrated that miRNA binding sites near the ends of transcripts are more effective[1,20]. Therefore, 3′-UTR shortening could be used as a global regulatory tool to increase the effectiveness of binding sites upstream of the cleavage point by shortening their distance from the transcript 3′ end. Indeed, repression by miRNAs was more effective in transcripts with shortened 3′-UTRs compared to those with longer 3′-UTRs[21].

Our study provides a mechanistic explanation for these findings, as most likely the binding site of the miRNA only affected the body and the tail was not affected by the miRNA regulation. Moreover, our findings have a high impact on previous miRNA expression analyses, as most miRNA studies evaluated their regulatory effect using cDNA microarrays, which were usually designed to hybridize to the 3′-UTR ends of transcripts. Our data indicate that such an analysis could be systematically biased and might overlook the actual decrease in "body" mRNA expression. We believe that our results will similarly impact additional fields, e.g., studies related to RNA binding proteins.

Overall, our study sheds new light on the transcriptome and calls for a new perspective on the post-transcriptional regulation of genes through their 3′-UTRs. Furthermore, the discovery of thousands of processed 3′-UTR RNAs uncovers a significant class of RNA with diverse potential biological properties.

## Methods

**Mice**. C57BL6/J mice were maintained and bred under specific pathogen-free conditions in the Hebrew University animal facilities. All mouse studies were performed under protocol MD-16-14863 approved by the Hebrew University Institutional Animal Care and Use Committee. All mice were used for experiments between ~8 and 12 weeks.

**Mouse T cell isolation**. T cells were isolated from C57BL/6J (WT) mice spleens with an EasySep™ Mouse CD90.2 Isolation Kit (#18951) according to the manufacturer's instructions (STEMCELL Technologies).

**Nuclear/cytoplasmic RNA extraction**. Two million isolated T cells were subjected to Nuclear and cytoplasmic RNA extraction using the Cytoplasmic and Nuclear RNA Purification Kit (Norgen Biotek, 21000) following additional step of an on-column DNA removal using DNase I (NEB-M0303S).

**qRT-PCR and cDNA preparation**. cDNA was synthesized using ProtoScript First Strand cDNA Synthesis Kit with oligo-dT primers (New England BioLabs Inc.; E6300L). qRT-PCR was then performed using QuantStudio 12K Flex Real Time PCR system with a Power SYBR green PCR Master Mix Kit (Applied Biosystems).
Reaction was performed as follows:

1. 50 °C 2 min, 1 cycle.
2. 95 °C 10 min, 1 cycle.
3. 95 °C 15 s -> 60 °C 1 min, 40 cycles.
4. 95 °C 15 s, 1 cycle.
5. 60 °C 1 min, 1 cycle.
6. 95 °C 15 s, 1 cycle.

Data were normalized to Human/Mouse endogenous control (UBC) and analyzed using the ΔΔCt model unless otherwise indicated. Each experiment was performed in triplicates and was repeated three times. Student's t-test was used with 95% confidence interval.

Lists of all primers used in these experiments are presented in Supplementary Tables 1 and 2.

**RACE analyses**. Total RNA was extracted with QIAzol Reagent (Qiagen; 79306). One microgram of total RNA was subjected to SMARTer® RACE 5′/3′ Kit (Clontech Laboratories; 634858) according to the manufacturer's protocol. Then, PCR was preformed to RACE-Ready cDNA using the kit forward universal Primer and reverse Gene-Specific Primers listed in the primers list. PCR products were purified from agarose gel using NucleoSpin Gel and PCR Clean-up Kit (Macherey-Nagel; 740609) and were sequenced using ThermoFisher 96-capillary 3730xl DNA Analyzer. See Supplementary Table 3 for primers list for 5′-RACE experiments.

**Northern blot analysis**. PolyA+RNA samples (0.5 μg) were denatured for 10 min at 65 °C in RNA loading buffer containing 40% formamide and 1.3 M formaldehyde, and then separated on 1.6% agarose gel containing 1.17 M formaldehyde in 20 mM MOPS, 2 mM Na-Citrate and 1 mM EDTA (pH 8.0). RNA was transferred to Zeta-Probe membrane (Bio-Rad) by capillary transfer in 10× SSC (1.5 M NaCl and 150 mM Na-citrate). For Actb probing specific riboprobes were used. For synthesis of the riboprobes a template containing the T7 promoter upstream to Actb antisense were PCR-amplified using oligonucleotides presented in the "primers list". In vitro RNA transcription of the riboprobe was done using 600 ng of the PCR template, 0.5 mM of each CTP, GTP, and UTP, 20 μM ATP, and 30 μCi [$^{32}$P]αATP. See Supplementary Table 4 for primers used to generate riboprobes.

**RNA-sequencing**. RNA was extracted from cells using QIAzol Reagent (Qiagen; 79306) and treated with DNase I (NEB-M0303S). Five micrograms of RNA for each sample was processed with the NEBNext Poly(A) mRNA Magnetic Isolation Module (NEB; E7490) and further processed with the NEBNext Ultra Directional RNA Library Prep Kit (NEB; E7420S) or 3′−3′-mRNA-Seq Library Prep Kit (lexogen; 015UG009V0211).

**5′- CAP RNA immunoprecipitation**. For preclearing, 4 μl mouse IgG1 anti-Flag (A2220; Sigma) was added to 40 μl Protein G Agarose beads (E3403; Sigma) suspended in 0.5 ml PBS for 1.5 h in tumbling 4 °C. After two washes with PBS, the anti-Flag-coated beads were added to the poly(A) selected RNA extract in 0.5 ml binding buffer (50 mM Tris, 150 mM NaCl, 0.5% Triton, and 1 μl Murine RNAse inhibitor (NEB-M0314S)) and incubated for 1.5 h in 4 °C. Precleared RNA samples were then subjected to 5′-CAP RNA immunoprecipitation using 5 μl (1:100 dilution) anti-Cap antibody (Merck anti-m3G-cap, m7G-cap antibody, clone H-20) and incubated O/N in 4 °C. Then, RNA immune complexes were incubated with 40 μl of Protein G Agarose beads for 4 h in RT following four washing steps using wash buffer (50 mM Tris, 150 mM NaCl, 0.1% Triton, and 1 μl murine RNAse inhibitor). Beads were then re-suspended with 100 μl elution buffer (wash buffer containing 3 μl proteinase K NEB-P8107S and 0.1% SDS) and incubated for 15 min at 65 °C. The elution step was repeated one more time. Finally, 800 μl QIAzol was added to the 200 μl eluted RNA.
Three replicates were used for this experiment.

**3′-RNA capping and pulldown**. RNA was poly(A) selected using NEBNext Poly(A) mRNA Magnetic Isolation Module (NEB-E7490S). The elution step was skipped at this point and instead poly(T) beads/RNA complexes were washed with 50 mM Tris-HCl in DEPC water and then subjected to "on beads" capping using biotinylated GTP with the Vaccinia Capping System according to the manufacturer's instructions (NEB-M2080S—15 μl DEPC water, 2 μl 10× Capping Buffer, 1 μl Vaccinia Capping Enzyme, and 1 μl Biotin-11-GTP (Perkin-Elmer; NEL545001EA)). Beads were then washed twice with the NEBNext Poly(A) mRNA Magnetic Isolation Module and proceeded with the RNA elution protocol. After elution, labeled RNA with biotinylated GTP was pulled down using Dynabeads MyOne Streptavidin C1 (Invitrogen; 65001) according to the manufacturer's instructions.
Three replicates were used for this experiment.

**Terminator phosphate-dependent TEX treatment**. For RNA-seq analysis DNase I-treated, poly(A) selected RNA was treated with Terminator 5′-Phosphate-Dependent Exonuclease (Epicentre; TER51020) according to the manufacturer's instructions. Reaction was deactivated and RNA was purified using RNA Clean-Up and Conc MICRO-Elute Kit (Norgen; 61000).

Three replicates were used for this experiment.

**Generation of pGL3 luciferase reporter constructs**. pGL3 luciferase reporter constructs were created by cloning of 3′-UTR sequences of *ST6GALNAC4* or *RAP1B* (WT/mutant) into the *Xba*I site located at 3′-UTR of pGL3-Control vector.

See Supplementary Table 5 for primers used for cloning and Supplementary Notes 1 and 2 for the sequences of the cloned 3′-UTRs.

**Transfection and α-amanitin treatment**. U2OS, HEK-293, and HEK-293T cell lines were obtained from ATCC and grown in Dulbecco's modified Eagle's medium, supplemented with 10% fetal calf serum, 100 units/ml penicillin, and 100 µg/ml streptomycin at 37 °C.

For transfection, HEK-293 cells were plated on 35 mm plates in 40% confluency and grown for 16 h.

Transfection of pGL3 luciferase reporter vector was performed using TransIT-LT1 transfection reagent according to thr manufacturer's instructions. Cells were collected for analysis 24 h after transfection.

Transfection of miRNA, synthetic miR-92a-3p, let-7, or negative control dsRNA was performed using TransIT-X2TM Dynamic Delivery System according to the manufacturer's instructions. Cells were collected for analysis 40 h after transfection. Three replicates were used for this experiment.

To inhibit RNA polymerase II transcription activity, U2Os cells were incubated with α-amanitin (10 µg/ml; Sigma) for 6 or 9 h at 37 °C.

All cell lines were tested for mycoplasma contamination using EZ-PCR Mycoplasma Test Kit (Biological Industries; Cat. No. 20-700-20).

**RNA-Seq analysis**. *Trimming and filtering of raw reads*: NextSeq basecall files were converted to FASTQ files using the bcl2fastq (v.2.15.0.4) program with default parameters.

*QC preprocessing*: Raw reads were inspected for quality issues with FastQC (v.0.11.2), and were quality trimmed at both ends to a quality threshold of 32. Adapter sequences were then removed using cutadapt (version 1.7.1), through the Trim Galore! interface (version 0.3.7), leaving only reads of length above 15 nt. The remaining reads were further filtered to remove very low quality reads, using the fastq_quality_filter (FASTX package, version 0.0.14), with a quality threshold of 20 at 90% or more of the read positions.

*Genomic mapping of RNA-seq data*: The processed FASTQ files were mapped (using TopHat, v.2.0.13)[22] to the human genome and transcriptome (hg19). Reads that after processing were left as a pair, as well as reads for which only one of the pair mates remained, were used for further analyses. Mapping allowed up to 2 mismatches per read, a maximum gap of 5 bases, and a total edit distance of 7.

**3′-end RNA-seq analysis**. *Trimming and filtering of raw reads*: NextSeq basecall files were converted to FASTQ files using bcl2fastq (v.2.17.1.14). Reads were screened and preprocessed similarly to the described above, an additional step of removing polyA sequences from the 3′ ends of reads was performed with cutadapt, using a 75-mer oligo-A sequence as the "adapter" and a minimal overlap of 2.

*Mapping and peak calling*: The processed FASTQ files were then mapped to the human transcriptome and genome using TopHat (v.2.0.13)[22], allowing up to 3 mismatches per read, a maximum gap of 5 bases, and a total edit distance of 8. Peaks were called using mapped reads from all samples, using findPeaks (Homer package, v.3.12)[23], with "-region -size 200 -minDist 250 -strand" parameters. Peaks found within transcripts or up to 5 kb downstream of promoters were included in a GTF file and processed by the DEXSeq package.

**Segmentation to body/tail using transcript read coverage**. Analysis was performed separately for each transcript/treatment (TEX, 3′-pulldown, CAP IP). First, the average read coverage for both treatment and control were calculated for overlapping 50 bp windows (in 20 bp offsets) along the coding regions and 3′-UTR (excluding introns). Coverage values were then transformed to log scale.

Next, we trained a linear regression model to correct for different transformations of the results in control and various treatments. For this, we considered all windows that overlap coding regions (from all transcripts), and fitted a linear regression model $Y = aX + b$ for each treatment, where $Y$ is the log-transformed set of coverages along the transcript for every treatment, and $X$ is the log-transformed coverages for control. This allowed us to overcome experiment-specific bias and infer the offset and ratio between mean read coverage (per window) in every treated condition and control. We then applied the linear model to yield the expected read coverage for each transcript along both coding and 3′-UTR windows, and deviations from the regression model were calculated.

Next, we trained an HMM[24, 25] with two internal states corresponding to "Body" and "Tail" regions, allowing a single transition from the "Body" to the "Tail" state. Each state was characterized with a Gaussian emission of continuous variables (with shared variance). The parameters of the model were optimized using the Baum–Welch (Expectation–Maximization) algorithm, using maximum likelihood estimations. Finally, the most probable Body/Tail cleavage point was identified for every transcript/treatment as the maximum likelihood transition point between the Body and Tail states, using the Viterbi algorithm.

**miRNA analysis**. For this analysis, a custom annotation defining the body and tail regions as independent transcripts was built based on the statistically significant cleavage sites found in the 3′-UTR of transcripts by our HMM model in HEK-293 cells treated with TEX ("TEX-HMM cleavage sites"). The custom annotation was based on the hg19 RefSeq annotation from UCSC. The end coordinate of the last exon of the conventional transcript was changed to the coordinate of the TEX-HMM cleavage sites. A new "tail" transcript was added to the annotation and defined as a single exon transcript with its start site defined as the TEX-HMM cleavage site plus 1 and its end as the end coordinate of the original transcript. The gene_id, gene_name, and transcript_id of the new "tail" transcript remained identical to the original transcript with the addition of "B" at the end. This custom annotation was then used as the annotation file for the remainder of the differential gene expression analysis, which used HTSeq-count for counting reads in features (using the "exon" as feature type, "transcript_id" as the id attribute, and the "intersection-nonempty" mode), and DESeq2[26] with default parameters for the differential gene expression analysis, comparing the data between miRNA and control transfections. Transcripts with perfect base pairing to a 7-mer miRNA seed were considered as the miRNA targets.

See Supplementary Note 3 for miRNAs sequences used.

**Data availability**. The sequencing data generated during the course of this study are available at the Gene Expression Omnibus (GEO) under accession GSE84068. The HMM software is available via github at https://github.com/eranroz/polyA. The data that support the findings of this study are available from the corresponding author upon request.

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

## Acknowledgements

We thank M. Bronstein for helping with the construction of the RNA libraries for sequencing, Y. Nevo for helping with the data analysis, S. Sarfati for assisting with graphics, T. Ben-Zvi for technical assistance and N. Friedman for comments on the manuscript. This work was supported by grants from the Israel Science Foundation (grant nos.1275/12 and 913/15), the Israel Cancer Research Fund (grant no. 13/726/RCDA), Marie Curie People (grant no. 322006), and the Concern Foundation. T.K. and E.R. were supported by a Marie Curie CIG grant no. 618327. A.S.-S., E.R., H.M., and T.K. are members of the Israeli Center of Excellence (I-CORE) for Gene Regulation in Complex Human Disease (no. 41/11) and the Israeli Center of Excellence (I-CORE) for Chromatin and RNA in Gene Regulation (no. 1796/12).

## Author contributions

Y.M. designed, performed research, analyzed data, and wrote the manuscript; M.B. and T.K. designed research, analyzed data, and wrote the manuscript; H.M. and A.S.-S. analyzed data and wrote the manuscript. E.R. analyzed data; L.C.-D., E.A.S., and L.A performed research.

## Additional information

**Competing interests:** The authors declare no competing financial interests.

