## [Peer Review File · Nature Communications]

PEER REVIEW FILE

Reviewers' comments:

Reviewer #2 (Remarks to the Author):

In this manuscript, Malka et al identify thousands of cytoplasmic RNAs that correspond to the distal part of mRNA 3'UTRs. They demonstrate that the 5' ends of these RNAs correspond to locations of APA sites, and claim that these RNAs arise by additional cleavage of transcripts. Knowledge of this class of mRNAs will be important for RNA quantification measurements, e.g. as shown in the present manuscript for miRNAs.

There are a number of errors present that need to be corrected:

- The plots in Figure 3b and c correspond to the plots in Extended Data Fig 10c. However, when I look at RAP1B WT body vs tail, there is a significant difference in their relative expression in Figure 3b but there is no difference in Extended Data Fig 10c. This cannot be correct. It appears that all data in Extended Data Fig 10c and d are normalized to the first sample (Body/WT) but this is not stated in the legend.
- Figure 4d, what are the horizontal lines – are they the average coverage across exons as in Fig 2b?
- The blue line in the plot of Figure 2g is not adequately described.
- The new discussion on Page 8 suggests that the cleavage takes place in the nucleus. However, the authors also show that the cleavage occurs post-transcriptionally. Normally, transcripts are thought to be exported from the nucleus as soon as transcription/processing is complete. Thus, do the authors propose that the mRNAs are retained in the nucleus for up to 9 hours (see Figure 3f) for cleavage at the APA? They have not provided any evidence that the canonical cleavage and polyadenylation machinery is responsible. The discussion on page 8 does not clarify the situation – instead, it is more confusing.

Minor comments:

- The authors still use “novel category of RNA” in the abstract to describe their work, despite the request of Reviewer 3 to change this.
- I didn't notice a reference to Extended Data Fig 4c in the text.
- Bottom line of page 6, first reference to Extended Data Fig 14b should be for Extended Data Fig 14a
- Extended Data Figure 4: The contrast in panel c should be adjusted – bands are not visible on a printout. In the caption, the first line should read “... Actb RNA (a-b)”

Reviewer #3 (Remarks to the Author):

The authors present a compelling study and a well-written manuscript. I have no comments.

Responses to Reviewers:

Comment 1: The plots in Figure 3b and c correspond to the plots in Extended Data Fig 10c. However, when I look at RAP1B WT body vs tail, there is a significant difference in their relative expression in Figure 3b but there is no difference in Extended Data Fig 10c. This cannot be correct. It appears that all data in Extended Data Fig 10c and d are normalized to the first sample (Body/WT) but this is not stated in the legend.

Response: There must be some confusion, as the plots in Figure 3b and c **do not** directly correspond to the plots in Extended Data Fig 10c.

In Figure 3b and c we present the relative expression of the body and tail fragments upon **TEX treatment**. Hence, the decrease in the tail PCR results indicates that (in some molecules) it is in a separated fragment compared to the body fragment.

The results in Extended Data Fig 10c, on the other hand, present the relative expression of the body and tail (and across the cleavage site) in **TEX-untreated** RNA. Importantly, the Gap PCR probe (over the mutated cleavage point) shows higher RNA levels following mutations to the PAS APA signal.

To avoid confusion (due to poor color selection on our behalf), we have now changed the colors of these bars (in Extended Data Fig 10c).

We thank the reviewer and now clarified this point in the Figure legends.

Comment 2: Figure 4d, what are the horizontal lines – are they the average coverage across exons as in Fig 2b?

Response: We are sorry for the unclear presentation. These horizontal lines indicated the RNA coverage in the Tail fragment, and following the Reviewer 2's comment have been changed to indicate average RNA coverage across coding exons, as in Fig 2b.

Comment 3: The blue line in the plot of Figure 2g is not adequately described.

Response: Following the reviewer's comment we have clarified the description to read: "Blue line shows meta-gene plot of exon annotations, suggesting that the observed dip is not due to exon-intron junctions".

Comment 4: The new discussion on Page 8 suggests that the cleavage takes place in the nucleus. However, the authors also show that the cleavage occurs post-transcriptionally. Normally, transcripts

are thought to be exported from the nucleus as soon as transcription/processing is complete. Thus, do the authors propose that the mRNAs are retained in the nucleus for up to 9 hours (see Figure 3f) for cleavage at the APA? They have not provided any evidence that the canonical cleavage and polyadenylation machinery is responsible. The discussion on page 8 does not clarify the situation – instead, it is more confusing.

Response: We thank the reviewer for this comment, and have revised this whole paragraph to better present our data interpretation:

“Our data suggest that cleaved RNAs are present in the cytoplasm more frequently than in the nucleus. These findings are supported by Furger and colleagues (ref. 16) who described that APA isoforms with shorter 3'-UTRs are more frequent in the cytoplasm. In addition, although it is accepted that cleavage and polyadenylation occur co-transcriptionally, our study, as well as others (refs. 17,18), demonstrates that cleavage and polyadenylation at APA sites may occur post-transcriptionally as well. These two observations could be explained by mRNA retention in the nucleus followed by additional cleavage and export to the cytoplasm. Indeed, cleavage and polyadenylation in the nucleus are presented in Prasanth et al. (ref. 17), describing that some transcripts (such as CTN-RNA) are retained in the nucleus and are cleaved post-transcriptionally upon stress conditions to produce new mRNA transcripts with shorter 3'UTRs. In addition, synthesized RNA fragments with wild-type or mutated PAS proximal APA sites were incubated with nuclear extract of HeLa cells. In this in vitro assay, cleavage occurred at RNA fragments containing wild-type APA but not in RNA fragments containing mutated PAS (ref. 18). Additionally, our observations could be also explained by cleavage and polyadenylation at APA sites in the cytoplasm. To date, there is no evidence of such mechanisms.”

Minor comments:

Comment 1: The authors still use “novel category of RNA” in the abstract to describe their work, despite the request of Reviewer 3 to change this.

Response: We have deleted this term and revised the sentence to read: "suggesting a widespread post-transcriptional phenomenon producing thousands of stable 3'UTR RNA tails that exist alongside their transcripts of origin".

Comment 2: I didn't notice a reference to Extended Data Fig 4c in the text.

Response: A reference to Extended Data Fig 4c was added (page 4, line 6).

Comment 3: Bottom line of page 6, first reference to Extended Data Fig 14b should be for Extended Data Fig 14a.

Response: Thank you. We corrected it.

Comment 3: Extended Data Figure 4: The contrast in panel c should be adjusted – bands are not visible on a printout. In the caption, the first line should read “... Actb RNA (a-b)”

Response: Contrast of panel c was adjusted and the caption corrected accordingly.